# Participatory Workflow Analysis of Newborn Genetic Screening (NBS) to Support Tools for Improved Follow-Up: Comparing the Use Case of Hemoglobinopathy Traits Across U.S. States

**DOI:** 10.3390/ijns11020040

**Published:** 2025-05-20

**Authors:** Peter Taber, Jennifer Baysinger, Sierra Daniels, Natalie Diaz-Kincaid, Amy Gaviglio, Jacob Ginter, Patrice K. Held, Emily Reeves, Virginia Sack, Jennifer Weaver, Karen Eilbeck

**Affiliations:** 1Department of Biomedical Informatics, University of Utah, Salt Lake City, UT 84115, USA; keilbeck@genetics.utah.edu; 2Oklahoma Department of Health, Oklahoma City, OK 73102, USA; jenniferxa@health.ok.gov; 3Iowa Newborn Screening Program, Iowa City, IA 52242, USA; sierra-daniels@uiowa.edu (S.D.); jacob-ginter@uiowa.edu (J.G.); 4Iowa State Hygienic Lab, Ankeny, IA 50023, USA; 5Newborn Screening Program, Division of Disease Control & Health Statistics, Washington State Department of Health, Shoreline, WA 98155, USA; natalie.diaz-kincaid@doh.wa.gov; 6Connetics Consulting, LLC, Minneapolis, MN 55417, USA; amy.gaviglio@outlook.com; 7Stead Family Children’s Hospital, Iowa City, IA 52242, USA; 8Oregon State Public Health Laboratory, Oregon Health Authority, Hillsboro, OR 97124, USA; patrice.k.held@oha.oregon.gov; 9Florida Newborn Screening Follow-Up Program, Division of Children’s Medical Services, Tallahassee, FL 32399, USA; emily.reeves@flhealth.gov; 10Newborn Screening Program, Wadsworth Center, New York State Department of Health, Albany, NY 12237, USA; virginia.sack@health.ny.gov; 11Maternal and Child Health Division, Indiana Department of Health, Indianapolis, IN 46204, USA

**Keywords:** newborn screening, workflow analysis, informatics, participatory design, hemoglobinopathy

## Abstract

Communication of newborn screening (NBS) results often fails to provide clear explanations of NBS screen results to parents. Understanding existing NBS workflows is vital for improving NBS follow-up. We sought to describe a diverse range of state NBS programs as a starting point for designing tools to improve NBS follow-up, using the example of hemoglobinopathy traits. At a workshop of the 2023 Association of Public Health Laboratories NBS Symposium, participants filled out a survey and modeled their state workflows. Salient features were extracted and synthesized by state. A subset of models was member checked. Representatives from 19 U.S. states participated in the workflow analysis. Mail was overwhelmingly relied upon to convey the results. NBS programs differed by point of first contact with parents and degree of reliance on third parties. A participatory approach is useful for the rapid preliminary documentation of existing NBS program diversity and opportunities and challenges to improve patient education and follow-up. Future work should broaden the analysis to additional entities or individuals, particularly parents and caregivers.

## 1. Introduction

For many parents or guardians of newborns, receiving a putative positive result from newborn genetic screening (NBS) may be their first encounter with genomic testing and genetic data—an unexpected and potentially life-changing event. Adverse psychosocial outcomes affecting parents receiving NBS results have long been a concern [1,2,3,4]. Informatics solutions, such as applications, web portals, and chatbots may have a role to play in improving the experiences of families receiving NBS results, by providing vital information at optimal points in the NBS workflow. For example, family education and navigation can be provided in part via SMS text on mobile phones, which are owned in some form by 97% of all U.S. households, and 95% of households with an annual income less than $30,000 [5]. However, informatics interventions require an understanding of the baseline workflows to ensure that novel tools provide the right information to the right individuals or organizations at the right time [6,7,8]. In the context of NBS, informatics tools face significant challenges due to the diversity of workflows used by NBS programs and the lack of research documenting those workflows. This study sought to characterize a diverse sample of state NBS program workflows in the U.S. as a starting point for planning novel NBS tools to improve parent and caregiver experience of the return of results.

In the United States, NBS samples are usually sent to a central laboratory (often a state’s public health laboratory), where they are screened for a variety of diseases [9,10]. In cases of life-threatening disease, when immediate treatment initiation is critically important, NBS programs may reach out to parents directly. Outside of such cases, the exact steps and extent of follow-up processes for actionable results are highly variable by state and disease. Some variability in NBS workflows is currently captured in data gathered by the Association of Public Health Laboratories (APHL)’s NBS Technical assistance and Evaluations Program (NewSTEPs) [11]. However, NewSTEPs data does not capture comprehensive information about follow-up procedures. Two APHL surveys investigating follow-up workflows show the differences at the state level of what is screened and what is reported for the case of hemoglobinopathies, one of the most common NBS results. For alpha-thalassemia, 93% of responding programs reported some alpha-thalassemia results, with over half using a two-tier screening algorithm. The recommendations for further follow-up varied, and included steps like confirmatory testing and/or referral to a hematologist [12]. In a second survey, all responding states reported some beta-thalassemia screening, with 20% using only one screen [13]. For this disease, follow-up is also exceedingly varied. State labs revealed results to some combination of follow-up teams, physicians, and parents directly.

Existing resources thus suggest great variability in NBS workflows and highlight the need to fill our knowledge gap around current practice. While exhaustive documentation of all state programs was not feasible in this study, high-level analysis of diverse NBS programs can help to characterize salient program features to guide future informatics interventions in designing novel tools to support the return of results, patient education, and follow-up. Important factors that are not clearly available in the existing studies include questions such as the following: In a given NBS program, is the parent notified of NBS results by the lab, a specialist, their primary care provider (PCP), or some other party? Are they notified over the phone, by letter, or some other channel? How does the program communicate with the PCP or specialist? To provide a starting point for designing tools to improve patient experience, this study sought to characterize the existing workflows used in diverse U.S. NBS programs, focusing on the common use case of hemoglobinopathies.

## 2. Materials and Methods

### 2.1. Study Overview

This was a qualitative study of state newborn screening workflows guided by existing work in participatory design (PD). PD is a design paradigm that seeks to prioritize the knowledge and experiences of organizational insiders as a basis for the redesign of work practices or the creation of novel tools [14,15].

### 2.2. Data Collection

To gain insight into existing state workflows for reporting hemoglobinopathy traits and carriers, a workshop was organized at the APHL’s NBS Symposium in October 2023. The workshop was created with an open call to Symposium attendees associated with, or knowledgeable about their state’s NBS program to attend and share information. The 1.5 h workshop began with a ten-minute presentation of the study objectives, which focused on eliciting state workflows for the return of trait carrier results for hemoglobinopathies. While the prevalence of sickle cell disease (SCD) was higher than that of other conditions in this group, the workshop focused on the broader category of hemoglobinopathies, with the aim to capture the diversity of workflows for the range of possible carrier/trait results. After instructions were provided, attendees were given time to complete a paper questionnaire about their programs, and an opportunity to construct visual models of their state’s workflow. Models were constructed using large-format poster paper, markers, and sticky notes. Attendees were asked to record the systems and mechanism of communication of each interaction. For example, the Laboratory Information Management Systems (LIMSs) may send an electronic message to the state genetic counselor (GC) group, or the pediatrician may call the parent. Attendees were free to use their own labeling system and nomenclature to represent what they knew about their state’s NBS workflow for hemoglobinopathy traits.

### 2.3. Data Analysis

The questionnaire responses were recorded in Excel. Visual workflow models were transferred to Lucidchart^TM^ (Lucid Software, Inc., South Jordan, UT, USA), and features of interest were recorded in another Excel spreadsheet. For the purpose of this analysis, “contact” with parents was defined as any communication of results or other information about NBS after a screen was performed. “Third parties” were defined as entities that are not either parent’s healthcare provider or state employees, and who are involved in contacting parents after an NBS screen (e.g., contracted genetic counselor services or specialty community-based organizations funded via grant mechanisms). Member checking, a technique where researchers involve participants in reviewing and validating their own data and interpretations, was conducted by providing digitized workflow diagrams and some clarifying questions via email to those workshop attendees who expressed interest in remaining involved in the study [16]. In keeping with the PD orientation of this study, rather than attempting to formalize the workshop results (e.g., by translating them into business process models [17]), this study emphasized creating diagrams that were accurate and coherent in the views of workshop attendees themselves. Participants were also enabled to make edits or comments directly in Lucidchart™ workflow diagrams to represent their programs as clearly as possible. Modifications to the results table and Lucidchart™ workflow models were then made to finalize them based on feedback from workshop participants, who were also invited to be co-authors.

## 3. Results

Representatives from 19 states shared their workflows for the return of hemoglobinopathy trait results in the workshop, both via questionnaire (*n* = 16) and the construction of large-format paper workflow diagrams (*n* = 19). Individuals from seven states opted to join the study and member check the findings. Individuals from two other countries (Greece and Sweden) also generously contributed insights, although those results are not included in this analysis. Of the participating U.S. states, fifteen were one-screen programs (with one dried blood spot (DBS) collected at 24–48 h; CA, DE, FL, IN, IA, KS, MI, NJ, NY, ND, OK, PA, TN, VA, and WI) and four were two-screen states (with one DBS sample collected at 24–48 h and a second routine DBS collected at 7–14 days; OR, TX, UT, and WA). In four states (DE, IA, OR, and WA), PCPs were reported as being the exclusive party contacting families, and in three states (NY, OK and WI) PCPs made contact with families in parallel with state follow-up program staff. Five of the participating states (CA, IN, MI, PA, and TN) used privately contracted services or grantee organizations in some capacity, for example, to provide genetic counseling. The communication of trait results overwhelmingly relied on USPS mail, with only one state indicating any use of email to relay results to parents. Mechanisms for PCPs communicating trait results to parents were generally not reported. Nine of the participating states (CA, IA, KS, NY, ND, OK, TN, UT, and WI) indicated that their programs have multiple contact points with parents with trait results (some attempts at contact could happen in parallel, e.g., if a PCP and public health nurse both attempted to contact parents at roughly the same time). Eight states (DE, IA, IN, MI, ND, OR, PA, and WA) reported that there was no direct contact between the state public health department and parents (in those cases, contact was mediated, e.g., by third parties or the family’s PCP).

Figure 1 demonstrates an example of an NBS results process provided by workshop participants and co-authors for Iowa (J.G. and S.D.). The figure describes a one-screen system in which the first contact with the family is made by the PCP, and genetic counseling is offered via referral from the PCP to state contracted NBS genetic counselors.

From the workflow models provided by workshop participants, we distilled some important ways in which NBS programs vary, and that may impact the design and implementation of tools to improve parent and caregiver experience. These features provide a useful high-level starting point for conceptualizing variability between NBS programs in future attempts to design tools for NBS follow-up and patient education. Figure 2 provides a Sankey diagram showing some of the key program features identified by this study that would impact the implementation of any informatics-supported NBS follow-up tool, along with the numbers of included states along each path. These features include whether the state has a one-screen or two-screen system; who makes first contact with parents (state, PCP, third party, or multiple entities or groups); the presence or absence of multiple contact attempts to reach parents; and the channel used in first contact with parents to communicate hemoglobinopathy traits. Other features that we extracted but are not displayed include all entities contacting parents throughout the full NBS results process and the presence or absence of confirmatory testing for suspected positives.

Finally, Table 1 provides a summary of the key workflow features extracted and synthesized from both the questionnaire and paper workflow diagram data for all participating states.

## 4. Discussion

This study collected data on the workflows used in the return of NBS results to parents via a workshop format. These data serve as a starting point for understanding what variability in the program design will be faced by informatics-supported improvements in follow-up and education, and as a guide to understand where key opportunities for intervention exist. The study seeks to support the creation of generalizable tools and implementation models that can be used to improve NBS results reporting by conveying clear information through accessible channels to parents or caregivers of newborns who are affected, or those who carry the screened traits. Salient aspects of NBS workflows across the programs studied provide guidance for considering how the future participatory design of novel tools to support NBS follow-up may be approached, including a minimal set of entities to be involved in the design processes.

One striking finding of the workshop data is that the majority of states provide results only by mail, with a small number using mail alongside other methods or relying only on other methods like phone calls. Some existing research suggests that channels of communication other than mailed letters may be preferred by some parents [18]. More than half of participating states reported multiple follow-up attempts with parents for hemoglobinopathy trait results, suggesting a significant administrative burden that could be reduced through the use of appropriate digital communication technology. Similar lessons apply to states that currently send educational materials via mail or fax to healthcare systems, which could potentially be replaced with communication that arrives at its target audience faster, is easier to access, and more easily archived.

There are a number of challenges with informatics-supported program improvements in NBS follow-up indicated by the workshop results. While all states reported having some form of follow-up team, there is inconsistency in whether the follow-up program is the first or only party to contact parents. Additionally, some states split responsibilities for aspects of follow-up activities between the state NBS program and contracted or grantee organizations (“third parties”). Depending upon a state’s protocol, parents and/or PCPs may be contacted by multiple different entities to complete the NBS process, creating additional complexity in interventions to provide appropriate follow-up education or facilitate communication with specialists. State programs that rely on first contact and communication of results by PCPs or other non-state entities may have less knowledge of the channels of communication used and the informational content provided to parents, highlighting the importance of the incorporation of provider perspectives in the design of any future interventions. All of these complexities reinforce that the design and implementation of future informatics tools will require a participatory approach that includes parents and caregivers, NBS programs, and healthcare systems, as well as contracted or grantee organizations in order to improve the chances of success [19,20,21].

Given the diversity of workflows across states and the reliance of NBS workflows on diverse entities in any given state, informatics-based solutions provide an important opportunity for standardization, by allowing low-cost and accessible mechanisms for carefully curated education for families. Such content can be strategically targeted at points in the workflow where families are likely to require support or where they have a chance of being lost to follow-up. Taking the example of Iowa’s NBS workflow (Figure 1), patient education could be provided at the time of reporting results to the state database, in preparation for the family’s visit with their PCP, and/or in preparation for a family’s visit to the hematology clinic. By mapping the NBS program workflow, opportunities for targeting education can be identified, as well as the specific content required by families (disease-related information or explanation of the need for hematology, etc.).

While this study provided a valuable view of the existing workflows and practices around NBS return of results for hemoglobinopathy traits, it has limitations. The study used a convenience sample of conference attendees; although the conference is a major symposium in the field of NBS and is well attended by the target professional group, data collection from organizations or individuals that do not attend the APHL NBS Symposium would be vital to establish a more thorough understanding of NBS processes, particularly around i) parent and caregiver perceptions and experiences of the return of results, and ii) the precise timing of the delivery of results if delivered by a party other than the state follow-up team. The study also relied on two forms of self-reported data collection, which may be prone to recall and other biases. Further work on NBS workflows should seek to expand the data sources and more rigorously establish patterns across NBS programs as a basis for future generalizable approaches to NBS process improvement. Finally, one issue that was not systematically captured by the workshop participant data was the presence or absence of confirmatory tests, which is likely to add complexity to the workflows described.

## 5. Conclusions

The objective of this study was to identify some of the salient ways in which U.S. NBS programs vary, with the ultimate goal of supporting the design of generalizable informatics tools to improve NBS follow-up for parents. The results document the diverse entities that contact parents at different stages in the NBS results reporting process, the channels of communication used, and whether single or multiple attempts are made to contact parents, as reported by workshop attendees at the 2023 APHL NBS Symposium. The results from our workflow analysis represent a level of detail not readily captured in the existing publicly available resources on NBS such as NewSTEPS. Future work should expand the data sources used to describe the existing NBS workflows; seek to clarify divisions of responsibilities between state programs and other entities involved in NBS results reporting and follow-up; and engage all stakeholders in the NBS process in participatory design to identify ideal functional requirements for informatics tools that can support improved NBS follow-up across a wide range of program configurations.

## Figures and Tables

**Figure 1 IJNS-11-00040-f001:**
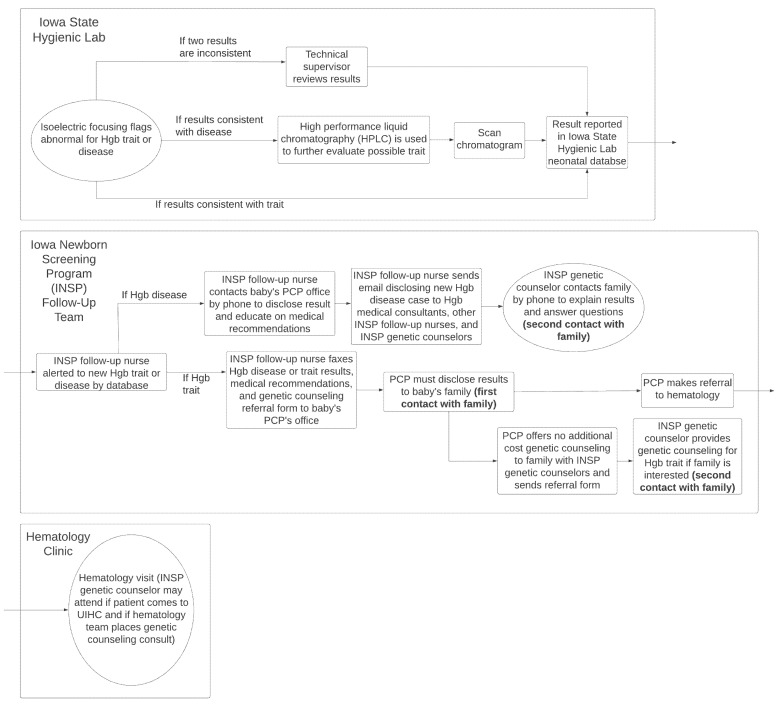
An example workflow for Iowa newborn screening (NBS) for hemoglobinopathy trait carriers (Hgb = hemoglobinopathy).

**Figure 2 IJNS-11-00040-f002:**
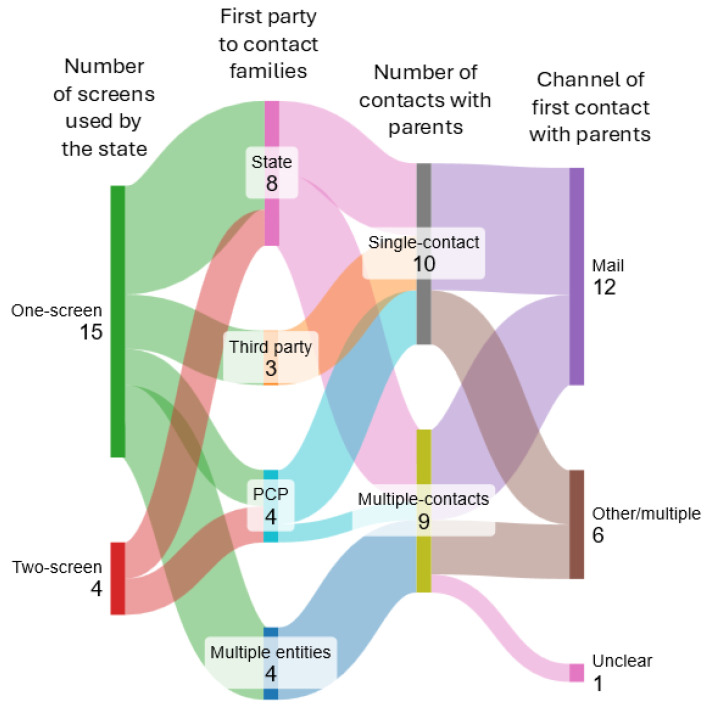
A Sankey diagram showing number of states with key features that would impact implementation of informatics-supported NBS follow-up tool using four questions: How many screens does the state perform? Which party makes the first contact with parents? How many times is contact attempted with a parent by any party for NBS return of results? And what is the first communication channel used by any party to contact parents for NBS return of results?

**Table 1 IJNS-11-00040-t001:** Extracted features of state newborn screening (NBS) for hemoglobinopathy trait carriers for 19 participating states. Blue records indicate member checking.

State	# Screens	State Contacts Parents in NBS Process	Party Making First Contact with Parents	Communication Channel(s) Used for First Contact with Parents	All Entities Indicated as Contacting Parents	Any Use of Third Party Indicated (e.g., Contractor or Grantee) in NBS Process
California	1	y	state	email, mail	state, contracted genetic counselor	y
Delaware	1	n	PCP	communicated by PCP	PCP	n
Florida	1	y	state	mail	state	n
Indiana	1	n	contractor/grantee	mail	contractor/grantee	y
Iowa	1	n	PCP	communicated by PCP	PCP, genetic counselor	n
Kansas	1	y	state	mail	state, PCP	n
Michigan	1	n	contractor/grantee	mail	contractor/grantee	y
New Jersey	1	y	state	mail	state	n
New York	1	y	state, PCP	mail	state, PCP	n
North Dakota	1	y	genetic counselor, PCP, specialist	phone	genetic counselor, PCP, specialist	n
Oklahoma	1	y	state, PCP	mail	state, PCP	y
Oregon	2	n	PCP	communicated by PCP	PCP	n
Pennsylvania	1	n	contractor/grantee	mail	contractor/grantee	y
Tennessee	1	y	state	mail	state, contractor/grantee	y
Texas	2	y	state	mail	state	n
Utah	2	y	state	mail	state, pediatrics/PCP	n
Virginia	1	y	state	mail	state	n
Washington	2	n	PCP	communicated by PCP	PCP	n
Wisconsin	1	y	state, PCP	unclear	state public health nurse, PCP	n

## Data Availability

Data collected in this study have been synthesized and published within the manuscript.

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
