# Peer review of "Participatory Workflow Analysis of Newborn Genetic Screening (NBS) to Support Tools for Improved Follow-Up: Comparing the Use Case of Hemoglobinopathy Traits Across U.S. States"

_2409-515X, 2025, doi:10.3390/ijns11020040_

Round 1

Reviewer 1 Report

Comments and Suggestions for Authors

Thank you for your well written and presented article. This paper aims to assess the pathways used to return the results of newborn haemoglobinopathy screens to families. The study requested participants attending a 2023 scientific meeting to outline the pathway's used in their state laboratory with final results for 19 US states. There are acknowledged limitations in that not all states are represented. None the less, this study serves as a good starting point to further explore outcomes and strategies, as well as potential benefits of AI as well as leaving the reader with a number of questions that it would be interesting and relevant to explore. 

I have no corrections. 

Reviewer 2 Report

Comments and Suggestions for Authors

Overall, this is a straightforward descriptive manuscript. It is a bit wordy and could be edited for conciseness. 

The authors' intended solution(s) for delivery of trait status it the use of some form of digital communication (..."replaced with communication that arrives at its target audience faster, is easier to access and more easily archived."). While these statements are true for many individuals US, it does not surmount the problem of lack or limited/changing access to digital solutions for a fraction of US families/individuals. How can such interventions be made available broadly (across levels of literacy or facility with technology)?

Can the authors provide a specific example of a proposed intervention, for example by using one State's or a hypothetical NBS program's workflow? This could be done in a figure.

Do the authors make any recommendations for uniformity (or decreased variability, at least) across NBS programs to help with the overall effort?

It would seem that sickle cell trait is the best example to use for this effort (instead of or in addition to alpha- and beta-thalassemia), given that newborn screening for hemoglobinopathies in the US is designed/intended primarily to screen for sickle cell disease.

Minor points:

P2: "While exhaustive documentation of all programs is not feasible...". Do the authors refer to feasibility for this study, or feasibility in general? Exhaustive documentation will be needed wherever an intervention, such as proposed in this manuscript, is planned.

P2: "Important factors that are not clearly available in existing include...". I believe a word is missing after "existing".

P2: "...workflows guided by exisintg work...". Existing is misspelled.

The use of the word "actors" throughout the manuscript is off-putting in this medical context. I suggest a different word, such as "entity" or "group" (or individual if appropriate).

P3. What does "member-checking" mean?

Lucidchart is a trademark and should be indicated as such and the manufacturer/commpany identified at first use.

Figure 1 is crudely drawn. The text should be fully enclosed by each shape.

Figure 2 is not labeled and does not have a legend.

In Figure 2 and elsewhere in the manuscript, does "multiple contacts" mean multiple attempts at contact, or multiple entitities/individuals contact the screened individual's family members?
